# A WaveGAN Approach for mmWave-Based FANET Topology Optimization

**DOI:** 10.3390/s24010006

**Published:** 2023-12-19

**Authors:** Enas Odat, Hakim Ghazzai, Ahmad Alsharoa

**Affiliations:** 1Electrical and Computer Engineering Department, Missouri University of Science and Technology, Rolla, MO 65401, USA; eo2fg@mst.edu; 2Computer, Electrical and Mathematical Science & Engineering (CEMSE) Division, King Abduulah University of Science and Technology (KAUST), Thuwal 23955, Saudi Arabia; hakim.ghazzai@kaust.edu.sa

**Keywords:** Generative Adversarial Network (GAN), deep learning, UAV, mmWave, FANET

## Abstract

The integration of dynamic Flying Ad hoc Networks (FANETs) and millimeter Wave (mmWave) technology can offer a promising solution for numerous data-intensive applications, as it enables the establishment of a robust flying infrastructure with significant data transmission capabilities. However, to enable effective mmWave communication within this dynamic network, it is essential to precisely align the steerable antennas mounted on Unmanned Aerial Vehicles (UAVs) with their corresponding peer units. Therefore, it is important to design a novel approach that can quickly determine an optimized alignment and network topology. In this paper, we propose a Generative Adversarial Network (GAN)-based approach, called WaveGAN, for FANET topology optimization aiming to maximize the network throughput by selecting the communication paths with the best channel conditions. The proposed approach consists of a WaveGAN model followed by a beam search. The former learns how to generate optimized network topologies from a supervised dataset, while the latter adjusts the generated topologies to meet the structure requirements of the mmWave-based FANET. The simulation results show that the proposed approach is able to quickly find FANET topologies with a very small optimality gap for different network sizes.

## 1. Introduction

Recent advances in the design of cost-effective embedded systems have broadened the use of unmanned aerial systems (UAS) in various applications, including aerial mapping, remote sensing, inspections, military operations, and more [1]. These systems often involve Unmanned Aerial Vehicles (UAVs), which can create Flying Ad hoc Networks (FANETs). FANETs are essentially impromptu mesh networks formed by UAVs self-organizing into groups, allowing collaborative work to accomplish missions. For efficient operation, these systems require the exchange of data among UAVs, relying on different communication technologies tailored to the specific requirements of the network quality of service (QoS). Furthermore, the growing field of the Internet of Drone Things (IoDT) [2,3] encourages the merging of FANETs with advanced communication methods. Technologies such as millimeter-wave (mmWave) communication [4] are particularly important in this context, catering to the substantial data demands of interconnected devices.

Although mmWave-based FANETs are a promising solution to achieve high-data-rate communication, there are several challenges that need to be addressed to maximize the potential benefits of this integration [5]. This is especially important considering that mmWave technology is characterized by its high bandwidth (10–300 GHz band), its sensitivity to the path loss effect, and signal attenuation due to the atmospheric effect and blockage, as well as a mandatory beam alignment [6]. Due to the short wavelength of the mmWave, multiple antennas could be packed together on the transceiver chips to steer narrow mmWave beams in different desired directions. Beam steering could be either performed in a mechanical or electronic way [7,8]. Mechanical beam steering relies on the physical manipulation of the antenna or its components to adjust the beam direction. This is typically achieved by rotating or tilting the antenna array mechanically, using devices such as motors. Although this method can be effective in aligning the beam with the target, it is generally slower due to the inherent limitations of physical movement. This slower response time, coupled with the potential wear and tear of mechanical parts, can be a drawback, particularly in applications requiring rapid directional shifts. In contrast, electronic beam steering employs phased array systems, where beam direction is controlled by electronically altering the phase of signals across multiple antenna elements. This approach eliminates the need for moving parts, leading to quicker and more precise beam adjustments. The ability to rapidly change beam direction through electronic phase variation offers significant advantages in high-speed communication scenarios and environments demanding swift reconfiguration. Furthermore, the absence of mechanical components in electronic steering systems enhances reliability and reduces maintenance needs, making it a more favorable choice for modern mmWave applications. With FANETs, the network topology is highly dynamic because of the mobility and limited resources, e.g., energy, of the flying units. Therefore, in FANETs based on mmWave, the UAV can establish a single-hop link with any other UAV under certain conditions. First, the direction of the antennas needs to be steered toward the best direction to establish a link with a neighboring node. Second, the distance that separates the two flying transceivers must not exceed the communication range to ensure seamless transmission. Therefore, due to increased path loss, it becomes necessary to establish multihop routes for data transmission between the source and destination.

Several previous works investigated the routing problem in FANETs [9] and mmWave networks [10,11]. However, these solutions may not always be applicable when both technologies are jointly deployed. In [12], the authors discussed the finding of optimal positioning of UAVs based on the best link quality of the mmWave beams. In [13], the problem of establishing the best beam pair to establish a direct link for each UAV pair in the fly mesh network is investigated. In this UAV mesh network, the optimization problem is formulated to maximize the Signal-to-Noise Ratio (SNR). Due to the complexity of the algorithm, a fast beam tracking method based on thresholding is proposed to find a suboptimal solution.

In many applications, such as network backhauling or ad hoc networks [13,14], orchestrating the mmWave network remains a challenging task. In fact, the beam alignment of the steerable antennas is required to establish links between any two nodes. Consequently, establishing a network topology that enables the exchange, dissemination, and collection of data across all these nodes necessitates the use of effective optimization techniques. Generally, such problems are classified as NP-hard, which are typically formulated as Mixed-Integer Programming (MIP) problems [15].

These problems are typically addressed using heuristic methods, which involve solving an iterative subproblem for each instance. In dynamic environments such as UAVs in FANETs, quick solutions are essential to determine optimized network topologies in response to updates at the network level. In mmWave-based FANETs, factors such as node failure or mobility are critical, as beam misalignment can lead to a complete transmission shutdown.

For many years, researchers have delved into routing problems across various applications, employing diverse methods with varying degrees of complexity. Deep learning algorithms for similar problems have recently been investigated as an alternative solution to find data routes in wireless networks due to the evolution of high-performance computing and the availability of huge amounts of data required to train models [16]. Once trained, deep learning models can provide solutions for the new network’s topology in a fast processing time. In [17], the authors discussed the use of deep learning to find the shortest route for data routing in wireless sensor networks. The work in [18] proposed the integration of Graph Neural Networks (GNNs) with deep reinforcement learning to solve the routing problem on graphs of different sizes and structures. In [19], a deep reinforcement learning technique has been proposed to dynamically maintain links for routing protocols in FANETs.

Recently, deep generative models have gained significant attention due to their ability to generate realistic and diverse outputs in various domains, including images, text, and audio [20]. For example, they are used for image synthesis, text generation, machine translation, dialogue systems, and drug discovery, among others. It is a class of deep neural network models that learns the underlying patterns and distribution of training data and generates new data samples that mimic the characteristics of the original data. They can be categorized into two main types: explicit density models and implicit density models [21]. Explicit density models provide a parametric specification of the data distribution by defining computationally tractable density functions, such as deep belief networks, or intractable density functions that use approximations, such as Variational Autoencoders (VAEs) [15]. In contrast, implicit generative models are a class of generative models that do not explicitly specify the data distribution. Instead, the aim is to define a stochastic process that can generate samples from the target data distribution. One prominent example of an implicit generative model is the Generative Adversarial Network (GAN) [22]. GANs consist of a generator network and a discriminator network that engage in an adversarial learning process. The generator generates synthetic samples, while the discriminator tries to distinguish between real and generated samples. Through this competition, the generator learns to produce samples that resemble real data, while the discriminator learns to improve its ability to differentiate between real and generated data.

GANs have been used in graph representation learning [23,24,25] and networking applications such as data enhancement and anomaly detection [26]. The motivation for using GANs lies in their ability to generate realistic and diverse data samples that can capture the complex and high-dimensional data distributions of the training data. Furthermore, GANs provide a framework for unsupervised learning, where the generator learns from unlabeled data without explicit labels or supervision. However, training GANs can be challenging due to issues like mode collapse, unstable training, and vanishing gradients. These challenges are addressed by developing novel architectures, loss functions, and training techniques to improve the performance of deep generative models [27].

This paper introduces the application of the deep generative WaveGAN model for optimizing network topology in dynamic mmWave-based FANETs, offering rapid and efficient solutions. In these networks, UAVs, equipped with steerable antennas, need to exchange data efficiently during activities such as hovering or area monitoring. The primary goal is to manage the activation of mmWave links between UAVs to maintain high-quality service. The key contributions of this paper are summarized as follows:We propose a novel solution adapted to mmWave-based FANETs using deep generative models, which takes as inputs the 3D locations of the UAVs and the channel and communication parameters of the potential mmWave links to determine the network topology providing the highest achievable total throughput.Training of the proposed Wave–Generative Adversarial Network (WaveGAN)-based model on multiple instances of dataset realizations, which consist of the network’s cost matrices. The model is trained using different configurations and network sizes. During testing, WaveGAN, followed by a fast beam search approach, rapidly generates topology solutions for UAVs located at random positions. This process determines the optimal orientation of the steerable antennas. The simulation results demonstrate that the proposed WaveGAN-based approach efficiently infers solutions with a small optimality gap, showcasing its effectiveness for various networks’ sizes and parameters.

This paper is organized as follows. Section 2 describes the system and channel models. Section 3 formulates the problem of determining optimized topologies for FANET based on mmWave. The proposed WaveGAN-based approach is presented in Section 4. The simulation results are provided in Section 5. Finally, the paper is concluded in Section 6.

## 2. System and Channel Models

Consider a FANET deployed over a subarea $\Omega \subseteq R^{3}$. This network consists of *N* UAVs, each located at specific coordinates $X_{i}=\left(x_{i},y_{i},z_{i}\right)$, where i=1,…,N. The primary objective of these UAVs is to exchange data among themselves. Operating in the mmWave band, each UAV is equipped with two directional antennas, one with a transmission gain $G_{i}^{T}$ and the other with a reception gain $G_{i}^{R}$, specifically designed for mmWave frequencies. UAVs function as airborne transceivers, and their operation must be optimized to maximize data transfer performance. The layout of this FANET based on mmWave is shown in Figure 1.

In this study, we assume that each UAV employs a fixed transmit power $P_{i}$. The goal is to establish a network topology for data transmission where each UAV adjusts and aligns its transmission and reception antennas to connect with two peer UAVs. This setup allows each UAV to both receive and relay messages to another single UAV at any given moment.

We assume that the UAVs utilize the mmWave spectrum for wireless data transmission. Compared with the standard muwave band, mmWave technology offers a significant amount of free spectra (i.e., approximately 100 GHz) with wide bandwidths, allowing for the achievement of very high data rates. However, highly directed links are needed to handle the increased path losses. Frequency Range 2 (FR2) frequencies extend beyond 40 GHz, and several applications are exploring different frequency ranges above 40 GHz, such as the 60 GHz and 73 GHz bands. However, several challenges exist that can significantly degrade signals, including the ability to penetrate solid objects and the Line-of-Sight (LoS) requirement. Moreover, the signal is impacted not only by the free-space path loss effect but also by atmospheric conditions including oxygen, vapor, and rain. In our study, we present findings specifically for frequencies ranging from 24 to 40 GHz. Additionally, we highlight that our proposed methodology is adaptable to other frequency ranges. This adaptability can be achieved by altering the channel model, which is a crucial step in the generation of datasets for different frequencies [28].

We denote by $h_{ij}$ the channel gain of the link between UAV *i* and UAV *j*. We assume that the overall transmission of the data through the whole network is continuous, and hence, it is relatively long compared with the channel coherence time. Therefore, we focus on the system’s performance based on the average statistics of the channel. Toward that, we consider the large-scale path loss effect in the channel gain that is expressed as follows:(1)$$h_{ij}\left(\Delta_{ij}\right)=\frac{1}{\sqrt{PL_{ij}\left(\Delta_{ij}\right)}},$$
where $\Delta_{ij}$ is the distance separating the UAVs *i* and *j* and is expressed as $\Delta_{ij}=\left|\right|X_{i}-X_{j}{\left|\right|}_{2}={\left({\left(x_{i}-x_{j}\right)}^{2}+{\left(y_{i}-y_{j}\right)}^{2}+{\left(z_{i}-z_{j}\right)}^{2}\right)}^{\frac{1}{2}}$, where ${\left||.|\right|}_{2}$ is the second norm distance. The term $PL_{ij}$ represents the path loss effect. Finally, we denote by ${\mathrm{SINR}}_{ij}$ the signal-to-interference-plus-noise ratio between UAVs *i* and *j*, which is expressed as follows:(2)$${\mathrm{SINR}}_{ij}=\frac{P_{i}G_{i}^{T}G_{j}^{R}{\left|h_{ij}\right|}^{2}}{I+N_{0}B},$$
where I accounts for the average external interference, $N_{0}$ is the power per frequency unit of an additive white Gaussian noise, and *B* is the channel bandwidth. In this context, the effect of interference on the mmWave band is ignored because all nodes employ directional antennas. Given that Line-of-Sight (LoS) links are established between two airborne nodes, an Air-to-Air (A2A) mmWave link connecting two UAVs is considered, and the path loss in dB is calculated as follows:(3)$$PL_{ij}^{A2A}=10n\log_{10}\left(\frac{4\pi f\Delta_{ij}}{C}\right)+L_{\mathrm{LoS}},$$
where *n* is the path loss exponent, *f* is the carrier frequency, *C* is the speed of light, and $L_{\mathrm{LoS}}$ is the average additional loss due to LoS link, with its value depending on the various environmental factors.

In this paper, we focus primarily on network backhauling; therefore, aspects of Air-to-Ground (A2G) communication are not covered in our study. For models and discussions related to the complexities of A2G communication, readers are referred to [29,30].

In addition, mmWave communication is subjected to atmospheric conditions. The attenuation effect in dB at a given distance $\Delta_{ij}$ is modeled by the International Telecommunication Union (ITU) as follows [31]:(4)$$PL_{ij}^{\mathrm{Atm}}=\left(L_{VO}\left(\Delta_{ij}\right)+L_{R}\frac{\Delta_{ij}}{1000}\right),$$
where $L_{VO}$ corresponds to the signal attenuation due to vapor water and oxygen, and $L_{R}$ is the attenuation due to rain [31]. Therefore, the total average path loss of mmWave links in dB can be expressed as follows:(5)$$PL_{ij}=PL_{ij}^{\mathrm{Atm}}+PL_{ij}^{A2A}.$$

## 3. Problem Formulation

To model the network topology optimization problem for the mmWave-based FANET system, we consider a weighted graph G=(N,E) of set N nodes and E⊂N×N edges. Each node i∈N in the 3D space corresponds to a UAV, while each mmWave link is represented by an edge going from node i∈N to node j∈N as $\epsilon_{ij}\in E$. We assume that there is at most one edge between any two nodes, the graph has no self-loops, and all edges are undirected. Hence, the mmWave-based FANET topology can be represented by a symmetric adjacency matrix, $A\in R^{\left|N|\times |N\right|}$, such that
(6)$$A_{ij}=\left\{\begin{array}{cc} 1 & \;\mathrm{if}\;\epsilon_{ij}\in E,\\ 0 & \;\mathrm{otherwise}.\; \end{array}\right..$$
where $A_{ij}$ represents the element in A. The impossible existence of a link between two nodes at given 3D locations is due to the short-range communication of the A2A mmWave link. Hence, we assume that if the ${\mathrm{SINR}}_{ij}\leq {\mathrm{SINR}}_{th}$, no transmission can occur, where ${\mathrm{SINR}}_{th}$ is a threshold indicating that below its value, the receiver cannot perfectly detect the signal.

Our objective is to determine an optimal FANET topology (i.e., alignment of the mmWave antennas) that provides the highest quality of service, i.e., the achievable data rate in our case, by selecting the links with the highest SINRs between different nodes. Therefore, we set the cost $c_{ij}$ of an edge $\epsilon_{ij}$ as follows: (7)$$c_{ij}=\left\{\begin{array}{cc} \frac{1}{{\mathrm{SINR}}_{ij}}, & \;\mathrm{if}\;\epsilon_{ij}\in E,\\ 0 & \;\mathrm{otherwise}.\; \end{array}\right.$$We define the cost vector of the edges, $C\in R^{\frac{\left|N|(|N|-1\right)}{2}}$, where each of its elements is $c_{ij}$. Additionally, we represent the adjacency matrix of the optimal FANET topology (also referred to as the topology matrix) by $A^{*}$, defining it as
(8)$$A_{ij}^{*}=\left\{\begin{array}{cc} 1 & \mathrm{if}\;\mathrm{the}\;\mathrm{optimal}\;\mathrm{FANET}\;\mathrm{topology}\;\mathrm{includes}\;\epsilon_{ij},\\ 0 & \;\mathrm{otherwise}.\; \end{array}\right..$$Consequently, we denote by $C^{*}\in R^{\frac{\left|N|(|N|-1\right)}{2}}$ the optimal cost vector containing the cost of the selected edges of the optimal FANET topology. The corresponding edges will constitute the active elements of the optimal adjacency matrix $A^{*}$.

Let us define the binary decision variable, $\gamma_{ij}$, as
(9)$$\gamma_{ij}=\left\{\begin{array}{cc} 1 & \;\mathrm{if}\;\mathrm{the}\;\mathrm{FANET}\;\mathrm{topology}\;\mathrm{includes}\;\mathrm{the}\;\mathrm{edge}\;\epsilon_{ij},\\ 0 & \;\mathrm{otherwise}.\; \end{array}\right.$$Therefore, we can formulate our optimization problem as Integer Linear Program (ILP) formulation for determining the optimal topology for the mmWave-based FANET as follows:(10)$$\begin{array}{cc} \underset{\gamma_{ij}\in E}{\mathrm{minimize}} & \;C_{\mathrm{tot}}=\sum_{i=1}^{\left|N\right|}\sum_{j\neq i,j=1}^{\left|N\right|}c_{ij}\gamma_{ij}\\ \;\mathrm{subject}\;\mathrm{to}\; & \sum_{j=1}^{\left|N\right|}\gamma_{ij}=1,\\ \; & \sum_{i=1}^{\left|N\right|}\gamma_{ij}=1,\\ \; & \sum_{i=1}\sum_{j=1}\gamma_{ij}\leq \left|S\right|-1,\\ \; & \gamma_{ij}\in \left\{0,1\right\} \end{array}$$
i,j∈N,S⊂N,2≤|S|≤|N|−2
where $C_{\mathrm{tot}}$ denotes the total cost of the FANET topology and S is the set of all topology configurations in G. In (Equation 10), the first two constraints indicate that each UAV has only one transmit and one receive steerable antennas and hence, only one ingoing and one outgoing link can be established, while the third constraint ensures that the FANET topology is connected.

The problem described in (Equation 10) is NP-hard and can be solved optimally through exhaustive search methods. Exact solvers like Concorde utilize cutting-plane methods combined with branch and bound techniques, solving linear programming relaxations to determine the optimal solution [32]. However, these methods are computationally intensive and not well suited for dynamic network topologies like those in FANETs. As an alternative, ILP problems can be approached using heuristic methods, which employ search algorithms to find an approximate, suboptimal solution.

## 4. Generative Adversarial Networks (GANs)-Based Solution

The flexibility of UAVs in FANETs suggests devising a rapid approach to determine an optimized FANET topology, considering the characteristics of the mmWave links. In this study, we propose a novel deep learning approach based on a generative model GAN to determine a network topology for the given 3D positions of the UAVs. During the training phase, the GAN learns to determine optimized values of $\gamma_{ij}$ based on a supervised dataset that we provide. The dataset includes random 3D positions of *N* UAVs along with their corresponding optimal network topology. Therefore, in the testing phase, for any set of *N* random UAV positions, the GAN-based method generates an optimized network topology, drawing from its training experience. Given that deep learning models are not always perfectly accurate, the topology generated by the GAN model is subsequently input into a low-complexity beam search algorithm. This step is crucial for quickly adjusting true negatives, false positives, and false negatives in the links, as well as finding a complete tour that meets the problem’s constraints. The steps of this proposed approach are outlined in Figure 2.

The GAN model was initially proposed by Goodfellow in 2014 [22]. It consists of two parts: the discriminator, *D*, and the generator, *G*, which compete in a minmax game. The generator learns to fool the discriminator by generating fake data that are similar to the real input using a random vector as input. The discriminator’s objective is to determine whether its input is fake or real. Despite the novelty of the GAN, the original model has several issues, including the mode collapse and the vanishing gradient problems, which may not be adapted to the investigated problem. To cope with these issues, to make the training more stable and to solve the problem of mode collapse, another GAN version called Wasserstein GAN (WGAN) is adopted in this study [33].

We propose a WaveGAN model to find the optimal network topology that solves the problem in (Equation 10) based on a modified version of the WGAN model. The architecture of the generator and discriminator is shown in Figure 3. The generator starts with a cost vector as input, which passes through a series of transposed convolutional layers (Conv2dTranspose), each followed by a rectified linear unit (ReLU) activation function. These layers are designed to upsample the input vector to a higher-dimensional space, eventually leading to the output, which is a generated network topology. The final layer of the generator uses a sigmoid activation function to output a probabilistic adjacency matrix (or heat map). On the discriminator side, the input is a network topology that undergoes a series of convolutional layers (Conv2d) which work to downsample the input to a lower-dimensional representation. Each convolutional layer is followed by a LeakyReLU activation function, providing nonlinearity to the process. Both the generator and the discriminator include self-attention layers (SN), which help the model focus on important features in the input data and model long-range dependencies and batch normalization layers (BN) to stabilize the learning process.

Unlike the traditional GAN, the proposed WaveGAN learns a mapping from the cost vector C to the optimal adjacency matrix $A^{*}$. Specifically, the input to the generator is the cost vector C representing the cost of the edges among the nodes. The output of the generator is the fake optimal adjacency matrix, $\tilde{A^{*}}$. The discriminator takes the optimal adjacency matrix $A^{*}$ and learns to score the output of the generator, $\tilde{A^{*}}$. This score represents how real or fake the discriminator considers the output of the generator. The model is trained using the loss function (Equation 11), which is a mix of adversarial loss of WGAN (Equation 13) and ℓ1 loss:(11)$$L=L_{\mathrm{adv}}+L_{\ell 1}.$$
where $L_{\ell 1}$ represents the difference between the generator output and the real data, effectively minimizing the impact of noise, outliers, and overfitting. It is expressed as follows:(12)$$L_{\ell 1}=∥G\left(C\right)-A^{*}∥+∥(G\left(C^{*}\right)-A^{*})∥.$$Also $L_{\mathrm{adv}}$ is expressed as follows:(13)$$\begin{array}{c} L_{\mathrm{adv}}=\min_{G}\max_{D}\underset{\tilde{A^{*}}∼P_{g}}{E}\left[D\left(\tilde{A^{*}}\right)\right)]-\underset{A^{*}∼P_{r}}{E}\left[D\left(A^{*}\right)\right]+\lambda {\left({∥\nabla_{\hat{A^{*}}}D_{w}\left(\hat{A^{*}}\right)∥}_{2}-1\right)}^{2}, \end{array}$$
where $P_{r}$ is the distribution of real data, $P_{g}$ is the model distribution of the generated data, and $\hat{A^{*}}$ is the interpolation between the optimal adjacency matrix, $A^{*}$, and the generated one, $\tilde{A^{*}}$, used in the computation of $\lambda {\left({∥\nabla_{\hat{x}}D_{w}\left(\hat{x}\right)∥}_{2}-1\right)}^{2}$, which is a gradient penalty to ensure the Lipschitz constraint and stabilize the model training [34].

After training, the result is a probabilistic adjacency matrix (or heat map). Each entry in this matrix corresponds to the probability that a certain edge belongs to the optimal topology of the network. To convert this matrix into a valid final topology, we use the beam search method [17]. The beam search is a heuristic search algorithm that uses breadth-first search with limited width (also called the beamwidth β). At each step, instead of exploring all possible edges to connect with the current β partial network topologies, it expands only the next β possibilities. The algorithm continues to run recursively until all nodes in the network are included in the best β candidate solutions (network topologies). The final network topology, T, is selected such that it has the highest probability, which is defined as $P\left(T\right)=\prod_{\epsilon_{ij}\in T}\tilde{A_{ij}^{*}}$.

In the context of mmWave FANETs, the network’s topology may undergo changes over time due to various factors like environmental conditions and the limited resources of the nodes. Our proposed learning-based approach, WaveGAN, is designed to be adaptive and robust in the face of these topology changes. By training WaveGAN on variable network sizes, either from scratch or through the use of transfer learning [35], it becomes capable of generating optimal solutions in real time when provided with the updated network topology as input to the trained model, as illustrated in Figure 2.

## 5. Results and Discussion

We examine a geographical area measuring 1000×1000 m${}^{2}$, where the maximum and minimum altitudes for UAVs are set at 110 m and 30 m, respectively. This defines the operational space as Ω=1000×1000×(110−30) m${}^{3}$. The minimum altitude for the UAVs is chosen to ensure Air-to-Air (A2A) communication, as lower altitudes might not guarantee reliable connectivity due to obstacles, particularly in urban areas. The rest of the system and channel model parameters are given in Table 1. The proposed model is trained for different FANET sizes using training datasets of size 1,000,000 samples, testing datasets of size 10,000, and validation datasets of size 10,000. The datasets are generated using the Concorde solver, where each sample consists of the cost matrix of the underlying network formed by elements $c_{ij}$ as input features and the optimal solution as a response. For each dataset instance, we randomly generate the 3D locations of *N* UAVs in Ω following a uniform distribution while considering an exclusion distance to avoid two or more UAVs being positioned in the same location. The proposed model is trained to minimize the cost function (Equation 11) using the Adam optimizer [36] with Adam hyperparameters $\beta_{1}=0.5,\;\beta_{2}=0.999$. The learning rates of the discriminator and generator are fixed at 0.0002, and the training is conducted using a batch size of 32. In the beam search, we use a beam size of 1280, since it gives the best model performance. The model is trained using NVIDIA Quadro RTX 5000 GPU.

Figure 4 plots the loss of the generator and discriminator over the iterations of the training process for a FANET of size 50. The figure clearly demonstrates the adversarial behavior of the employed generative model. As previously mentioned, one of the main challenges that GAN models face is unstable training [27]. We handled this in our work by using the gradient penalty in the training loss function. We can also see that both losses stabilize to values near zero after a certain number of iterations, around 20,000. To further validate the convergence of the model, in Figure 5, we plot the mean squared error (MSE) between the ground truth (i.e., the optimal topology obtained by Concorde) and the predicted topology generated by the WaveGAN generator during model training. We notice that the MSE decreases at the beginning of the training until it stabilizes at a value near 0.01. This shows that the model is learning to find many true positive edges at the end of the training.

During the testing time, we measure the performance of the proposed model and compare it with that of the optimal solution across various network sizes. To this end, we compute the optimality gap, which corresponds to the ratio of the generated cost length ${\hat{C}}_{\mathrm{tot}}^{t}$ relative to the optimal length $C_{\mathrm{tot}}^{t}$ over the testing dataset. It can be computed as $\frac{1}{T}\sum_{t=1}^{T}\left({\hat{C}}_{\mathrm{tot}}^{t}/C_{\mathrm{tot}}^{t}-1\right)$, where *T* is the size of the testing dataset (*T* = 10,000). We also measure the computation time needed to find the optimized topology for Concorde (Optimal) and the proposed WaveGAN-based approach (Generator + Beam search). In Table 2, we provide the average results of mmWave-based FANETs with N∈{8,10,15,20,25,50}. It is shown that the proposed approach is able to determine solutions with a very low average optimality gap. In fact, after training, the generator can find, for any given 3D positioning of the UAVs, a network topology that is close enough to the optimal one. This network topology is afterward tuned by the beam search to correct the remaining errors (adjusting true negative, false positive, and false negative edges). For a small network size, such as N=8 and N=10, the optimality gap is, respectively, 0 and 0.2%; however, this optimality gap increases with the size of the FANET due to the fact that more realizations can occur and the generative model finds more difficulties to determine topologies close to the optimal ones. Nevertheless, the optimality gap remains very low (less than 7% for N=50). Additional training and larger datasets might be required to improve the performance of the model for larger network sizes. Although the training may last for a long period of time, once trained, the model can quickly find suboptimal solutions in significantly shorter computational times when compared with the Concorde solver. This makes the proposed approach very adaptive to dynamic networks such as FANET and helps find new topologies whenever an update occurs in the network.

Figure 6 displays examples of optimal topologies alongside the corresponding predictions for identical 3D UAV positions. It includes two types of predicted topologies: one generated by the WaveGAN model and the other modified post beam search for N=10 and N=20. This allows for a visual comparison, demonstrating how the WaveGAN generator creates network topologies nearly matching the ground truth, albeit with small inaccuracies. These errors can be corrected through the use of a rapid beam search algorithm, thus ensuring the final network structure adheres to the constraints of the optimization problem.

Finally, in Figure 7, we consider the scenario where a node is disseminating a message over the whole network following multihop links according to the obtained topology. We measure the achievable rate as the minimum rate over all the active links of the FANET. We average the obtained results over 2000 realizations, where at each realization, we place 20 UAVs randomly at different altitudes ranging from 30–45, 30–55, …, 30–105, as depicted in the *x*-axis of Figure 7. We also compare our proposed approach with the achievable rate obtained on the optimal topology (Concorde) and two other topologies obtained using a greedy algorithm and a nearest neighbor approach with 2-Opt heuristics (NN+2 Opt.) [38]. At each step, the greedy algorithm adds to the network topology the neighboring node with the highest edge probability and terminates when all nodes are added to the network. First, we notice that, for all approaches, the achieved throughput slightly decreases by increasing the altitude. This is because the distance between the UAVs increases, which impacts the mmWave path loss. On the other hand, we can see that the proposed approach achieves close results to the optimal solution with an average optimality gap of 6.25%, while the other approaches exceed 30%.

## 6. Conclusions

In this paper, we investigated the network topology optimization problem for a mmWave-based FANET with UAVs equipped with steerable antennas. We proposed a novel solution based on generative models where a trained WaveGAN followed by a beam search algorithm were developed to determine suboptimal network topologies meeting the requirements and constraints of the network. It was shown that the proposed approach is computationally complex during training but, once trained for a given network size, it is able to quickly find suboptimal topologies with a very low optimality gap for any 3D positions of the flying units. The rapidity of the method at testing is very adaptive to dynamic FANETs, where updates occur frequently.

At this stage, we investigated a type of FANET with specific conditions (two antennas with fixed 3D positions for UAVs) to test and validate this novel GAN-based approach to solve an NP-hard problem. As future work, we will focus on alleviating the training complexity of the generative model so as to explore more complex communication systems and network architectures, e.g., more antennas per unit and mobile UAVs with fixed trajectories. Additionally, our study will take into account various antenna-related factors, such as the estimation of antenna gain and the analysis of antenna polarization.

## Figures and Tables

**Figure 1 sensors-24-00006-f001:**
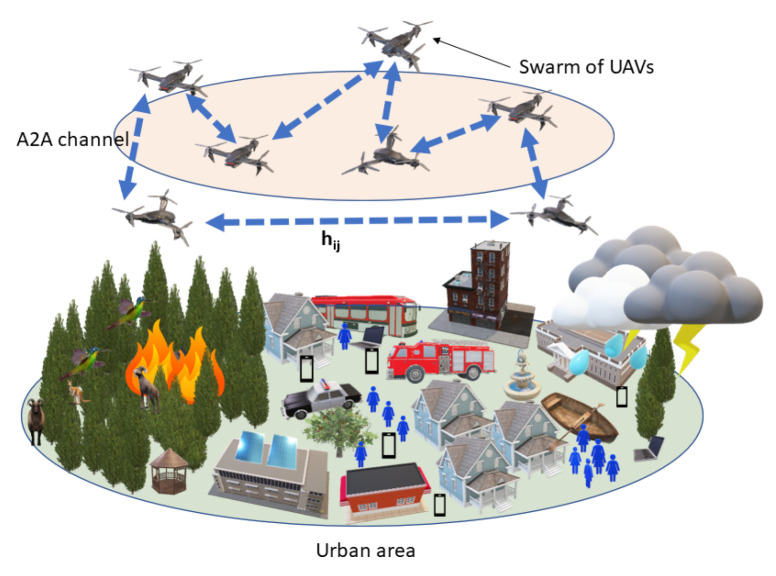
A mmWave-based UAV equipped with beam-steerable antennas forming a FANET.

**Figure 2 sensors-24-00006-f002:**
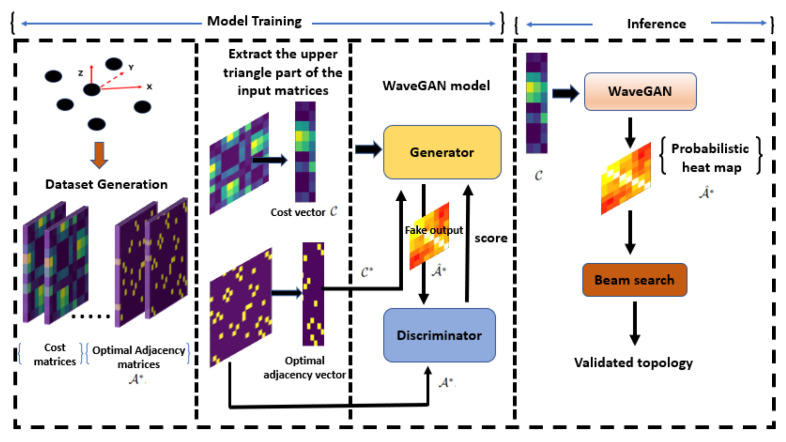
System flow: The first step is to randomly generate *M* training graphs, represented by the 3D UAV positions and the corresponding cost vector C. After that, the Concorde solver is used to find $A^{*}$, which is then used to compute $C^{*}$. The second step is to train the WaveGAN model. The input to the generator is C, and it outputs the fake topology matrix, $\tilde{A^{*}}$, which is jointly fed with $A^{*}$ to the discriminator. During the inference step, starting from a given random input C, the generator determines a network adjacency matrix $\tilde{A^{*}}$, which is then fed into the beam search to validate the final topology.

**Figure 3 sensors-24-00006-f003:**
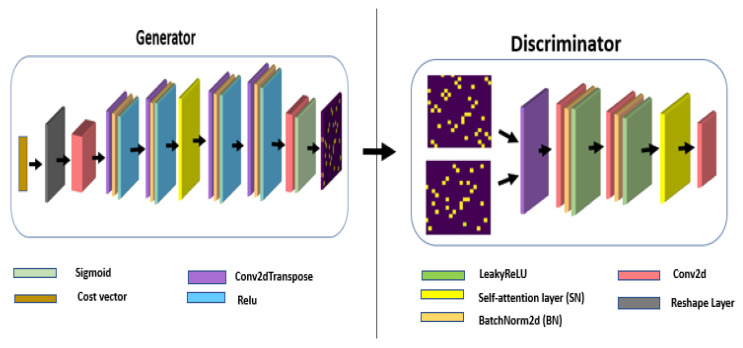
The proposed WaveGAN model architecture: The generator and discriminator consist of several layers of convolution, normalization, and activation. The strides and sizes of the convolution filters used at each layer are variables and depend on the size of the input graph.

**Figure 4 sensors-24-00006-f004:**
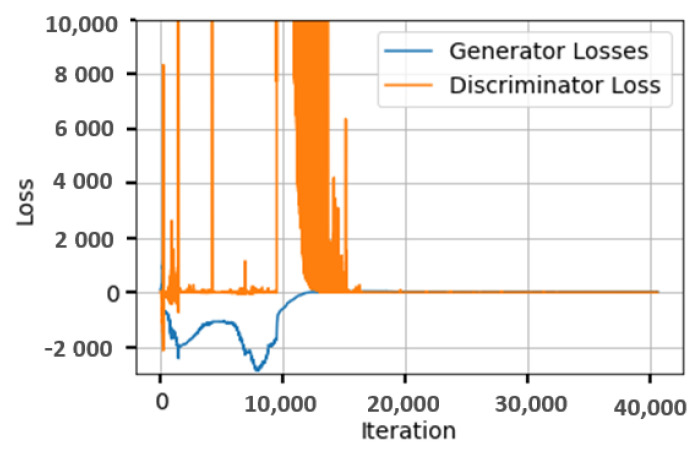
The loss functions of the generator and the discriminator of the proposed generative model over training.

**Figure 5 sensors-24-00006-f005:**
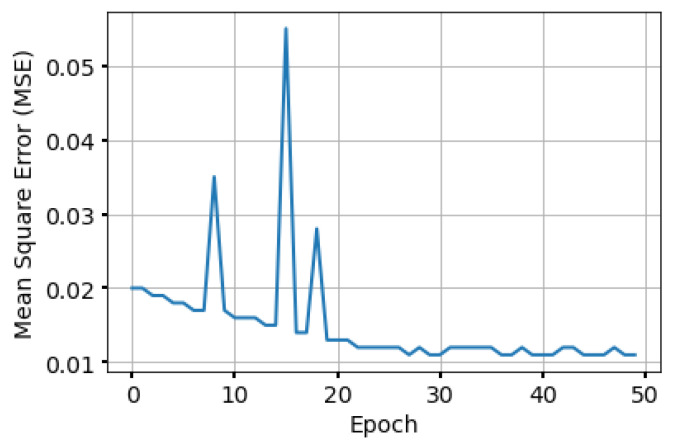
The MSE achieved by the proposed model during training.

**Figure 6 sensors-24-00006-f006:**
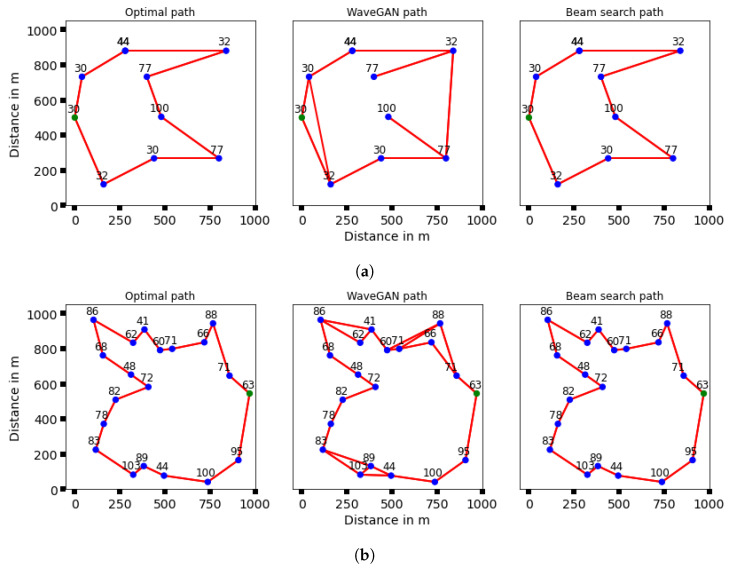
Examples of our method compared to other benchmarks for different number of UAVs (**a**) 10 UAVs, and (**b**) 20 UAVs. The optimal topology obtained by Concorde shows to the left, the topology generated by the WaveGAN generator given in the middle, and the topology generated after the beam search shows to the right. The number beside each node represents the height of the UAV.

**Figure 7 sensors-24-00006-f007:**
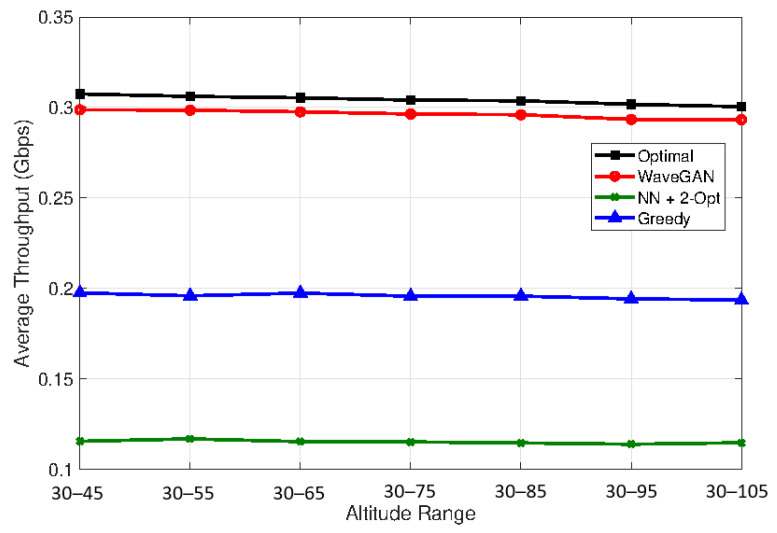
Monte Carlo simulation: achievable throughput versus different UAV altitudes. Comparison between the proposed approach and other suboptimal algorithms.

**Table 1 sensors-24-00006-t001:** Simulation parameters.

Parameter	Value	Parameter	Value
$f_{c}$ [GHz]	30	*B* [MHz]	500
$N_{0}$ [dBm]	−174	*n*	2
$L_{\mathrm{LoS}}$ [dB]	1	$L_{O}+L_{V}$ [dB/km]	14.25 $\frac{\Delta_{ij}}{1000}+0.01625$
$L_{R}$ [dB/km]	16.27	${\mathrm{SINR}}_{th}$ [dB]	−5 dB
$G_{i}^{T}$ [dBi]	37	$G_{i}^{R}$ [dBi]	37 ^1^

^1^ We assumed a specific antenna gain value for the purposes of model evaluation. It should be noted that a typical mmWave antenna with a 10-degree beamwidth usually demonstrates a gain of less than 30 dBi. For more comprehensive discussions about typical antenna gains across different applications, readers are referred to [37].

**Table 2 sensors-24-00006-t002:** The optimality gap and computation time of the WaveGAN-based approach compared with the optimal solution for different network sizes.

Problem	Concorde Solver	WaveGAN-Based Approach	
	**Time (s)**	**Opt. Gap**	**Time (s)**
N=8	102	0%	0.6
N=10	126	0.2%	0.64
N=15	210	3%	0.83
N=20	312	6.4%	1.04
N=25	612	6.5%	1.08
N=50	2112	6.97%	1.8

## Data Availability

Data are contained in this article and are available at: https://github.com/Odatems/WaveGAN-Repo-Remote.

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
