# Peer review of "A WaveGAN Approach for mmWave-Based FANET Topology Optimization"

_sensors, 2023, doi:10.3390/s24010006_

Round 1

Reviewer 1 Report

Comments and Suggestions for Authors

(1)"The article has many redundant sentences: In the Introduction section, the paragraphs from lines 86 to 104 and 114 to 138 that introduce deep models have many repetitive and redundant sentences. Please revise the above content, and carefully review the entire document."

(2)"There seems to be an issue with the layout of Figure 2 in the article."

(3)"The WaveGAN model architecture diagram represented in Figure 3 is not clear, and the article does not provide a specific explanation. Please make revisions."

(4)"In the field of millimeter-wave communication applications, beam search methods have been used in previous work. The beam search method proposed in the article seems to lack a detailed explanation of its specific innovative points. Please describe in detail the innovation of this method in the optimization of network topology in FANETs."

Comments on the Quality of English Language

The description of the related methods should be more concise.

Author Response

(1)"The article has many redundant sentences: In the Introduction section, the paragraphs from lines 86 to 104 and 114 to 138 that introduce deep models have many repetitive and redundant sentences. Please revise the above content, and carefully review the entire document."

Thank you for your comment. The paper has been revised as per advised.

(2)"There seems to be an issue with the layout of Figure 2 in the article."

Thank you for your comment. The layout has been improved in the revised version.

(3)"The WaveGAN model architecture diagram represented in Figure 3 is not clear, and the article does not provide a specific explanation. Please make revisions."

The following description has been added to the revised manuscript.

“The generator starts with a cost vector as input, which passes through a series of transposed convolutional layers (Conv2dTranspose), each followed by a rectified linear unit (ReLU) activation function. These layers are designed to upsample the input vector to a higher-dimensional space, eventually leading to the output, which is a generated network topology. The final layer of the generator uses a sigmoid activation function to output a probabilistic adjacency matrix (or heat map). On the discriminator side, the input is a network topology that undergoes a series of convolutional layers (Conv2d) which work to downsample the input to a lower dimensional representation. Each convolutional layer is followed by a LeakyReLU activation function, providing non-linearity to the process. Both the generator and the discriminator include self-attention layers (SN), which help the model to focus on important features in the input data and to model long-range dependencies, and batch normalization layers (BN) to stabilize the learning process.”

(4)"In the field of millimeter-wave communication applications, beam search methods have been used in previous work. The beam search method proposed in the article seems to lack a detailed explanation of its specific innovative points. Please describe in detail the innovation of this method in the optimization of network topology in FANETs."

Thank you for your valuable comment. The beam search algorithm, as introduced in [15], is a widely recognized method in various search-based problem-solving contexts, notably in the field of natural language processing. Its popularity stems from its ability to strike a balance between the computational complexity of greedy-based solutions and those based on combinatorial approaches. In this paper, the application of the beam search algorithm is specifically employed to validate the final topology of the network. This is crucial as the solutions generated by the GAN may include extraneous edges, which are not part of the desired final topology.

Comments on the Quality of English Language

The description of the related methods should be more concise.

Thank you for your comment. The description has been improved in the revised version of the manuscript.

Reviewer 2 Report

Comments and Suggestions for Authors

1.   1. Please explain how the beam of the antenna can be steered.

2. It would be helpful to include a comparison Table for the proposed model and those exit in the literature.

3. Is the model scalable to operate at another mmWave frequency, eg. 60 GHz?

4. Has the polarisation of the antenna been considered?

5. How the antennas gain has been estimated?

Comments on the Quality of English Language

The English language requires minor editions.

Author Response

  1. Please explain how the beam of the antenna can be steered.

Beam steering could be either done in mechanical or electronic way.  In a mechanical based approach, the mmWave antenna is usually steered through phased array systems. These systems consist of multiple antenna elements, where the phase of the signal at each element is controlled to steer the beam in different directions. By adjusting the phase shift across these elements, the antenna can dynamically direct the beam towards a specific receiver, improving signal strength and reliability. In the electronic case, mmWave systems often use electronic beam steering, which is faster and more reliable. We have included this explanation in the revised version and we have cited two references [7] and [8].

  1. It would be helpful to include a comparison Table for the proposed model and those that exist in the literature.

In our comprehensive review of existing literature, we did not identify any studies that address the specific problem we are tackling in the exact manner presented in our work. Given this context, to provide a meaningful evaluation and benchmark for our methods, we have opted to compare them with the techniques illustrated in Figure 7 of the paper. These methods, while not identical to our approach, share certain conceptual similarities and objectives, making them the most suitable candidates for comparison.

  1. Is the model scalable to operate at another mmWave frequency, eg. 60 GHz?

Thank you for your valuable feedback. Indeed, our solution model is designed with scalability and adaptability in mind, allowing it to function efficiently across various mmWave frequencies. By transitioning to a new frequency band, such as 60 GHz, the channel model may differ [28]. To address this, we propose employing transfer learning techniques. This approach involves adapting our already developed model to the specific characteristics of the new frequency band by training it on a dataset that reflects the unique aspects of the new channel model. This method ensures that our model remains robust and effective, even as it operates within the varying conditions presented by different mmWave frequencies. We have added this justification to the manuscript and cite reference [28].

  1. Has the polarization of the antenna been considered?

Thank you for your feedback. The specific specifications and detailed design aspects of antennas are not the primary focus of our current work. Our approach has been developed based on an abstract representation of the antennas. This abstraction allows us to concentrate on broader system-level solutions and performance assessments, rather than delving into the issues of individual antenna designs. This approach enables us to generalize our findings and apply our solutions irrespective of the specific antenna characteristics. However, we recognize the importance of antenna specifications in practical applications and anticipate that future work could integrate more detailed antenna models to further refine and validate our solution in specific contexts. We have added this this comment as a future work.

  1. How has the antenna gain been estimated?

 In this work, we operate under the assumption that the antenna gains are predetermined and provided. As such, the estimation of antenna gain values falls outside the primary focus of our research. This approach allows us to concentrate on other critical aspects of the system's performance and optimization. We acknowledge, however, that antenna gain is a significant factor in practical implementations, and its estimation could be an important area for future studies. For the purposes of our current research, our findings and conclusions are based on these given antennas gain values. We have added this comment as a future work.

Comments on the Quality of English Language

The English language requires minor editions.

Thank you for your comment. The English of the paper has been revised as advised.

Reviewer 3 Report

Comments and Suggestions for Authors

In this paper, a method based on generative adversarial network (GAN), called WaveGAN, is proposed to study the network topology optimization problem in dynamic millimeter wave (mmWave) flying Ad-hoc networks (FANETs). A deep generation model is used to determine the network topology to achieve the maximum attainable throughput. The study used training sets and test sets, and combined with a low-complexity beam search algorithm to adjust the generated topology.

However, this manuscript should be revised before being considered for publication. Some comments are listed as follow.

1)       This article proposed an unmanned aerial vehicle (UAV) air-to-air (A2A) channel model, considering the A2A mmWave link and atmospheric conditions. However, the segmented approaches to obtain the path loss are discussed in many papers. For example, in the paper titled 'Channel modeling for UAV-to-ground communications with posture variation and fuselage scattering effect', path loss takes into account air and ground characteristics. It is suggested that the author compare more UAV path loss researches.

2)       How to solve the problem of computational complexity during the training process if the size of the network is not given?

3)       The additional training and larger data sets may be needed for larger networks. Could the author specify what additional conditions are required?

4)       Although mmWave communication has a wider bandwidth, the propagation process and energy attenuation of signals with different frequency points are significantly different, which can be found in ‘A UAV-aided real-time channel sounder for highly dynamic nonstationary A2G scenarios’. Did the author consider the frequency non-stationarity of millimeter wave channels?

5)       The explanation of Figure 5 is not rigorous enough. Mean squared error (MSE) does not decrease with the increase of Epoch all the time. The author should explain the three sudden increases in MSE while the Epoch is less than 20.

6)       What did the numbers next to the UAV nodes in Figure 6 mean?

7)       In the third paragraph on page 3, ‘More specifically, it addresses the …… by utilizing deep-generative models’, the meaning here is basically the same as the previous sentence. The author should check the full manuscript to avoid repeated expressions.

Comments on the Quality of English Language

Minor editing of English language required.

Author Response

In this paper, a method based on generative adversarial network (GAN), called WaveGAN, is proposed to study the network topology optimization problem in dynamic millimeter wave (mmWave) flying Ad-hoc networks (FANETs). A deep generation model is used to determine the network topology to achieve the maximum attainable throughput. The study used training sets and test sets, and combined with a low-complexity beam search algorithm to adjust the generated topology.

However, this manuscript should be revised before being considered for publication. Some comments are listed as follow.

1)       This article proposed an unmanned aerial vehicle (UAV) air-to-air (A2A) channel model, considering the A2A mmWave link and atmospheric conditions. However, the segmented approaches to obtain the path loss are discussed in many papers. For example, in the paper titled 'Channel modeling for UAV-to-ground communications with posture variation and fuselage scattering effect', path loss takes into account air and ground characteristics. It is suggested that the author compare more UAV path loss researches.

Thank you for your comment. The focus of our work is specifically on backbone networks. Consequently, aspects of air-to-ground communication are not addressed within the scope of our research. While air-to-ground communication is undoubtedly a significant area in the broader field, it falls outside the purview of our current research focus. We have cited this paper in the revised version (i.e.,  [29]).

2)       How to solve the problem of computational complexity during the training process if the size of the network is not given?

Thank you for your comment. In our approach, the initial training of the model is conducted using datasets that correspond to a specific network size, which is a known and defined parameter. To adapt the model to new network sizes, we employ transfer learning techniques [35]. This method allows us to efficiently adjust the model to accommodate different network scales without the need to start the training process from scratch. In the fine-tuning stage, the computational complexity is significantly reduced since we leverage the pretrained model. This approach not only enhances the adaptability of our model to various network sizes but also ensures a more resource-efficient training process. Reference [35] has been added to the text of the revised version to emphasize this point.

3)       The additional training and larger data sets may be needed for larger networks. Could the author specify what additional conditions are required?

Thank you for your feedback. Indeed, to train our model for larger network sizes, the primary requirement is a dataset comprising networks of the new, larger size. This dataset will enable the model to learn and adapt to the characteristics and dynamics specific to larger networks. By focusing on datasets that reflect the targeted increase in network size, we can ensure that our model remains accurate and effective when scaled up, without the need for additional complex adjustments.

4)       Although mmWave communication has a wider bandwidth, the propagation process and energy attenuation of signals with different frequency points are significantly different, which can be found in ‘A UAV-aided real-time channel sounder for highly dynamic nonstationary A2G scenarios’. Did the author consider the frequency non-stationarity of millimeter wave channels?

Thank you for your comment. It's important to note that this paper does not focus on Air-to-Ground (A2G) communication. Regarding the non-stationarity of the channels, our current work does not explicitly address this aspect. However, we acknowledge that the variation in channel characteristics, including those due to different channel frequencies, is an important factor in wireless communication. We believe that the channel models presented in this paper have the potential to be adapted to account for these variations. By incorporating new data that corresponds to different channel conditions and frequencies, our solution could be tailored to handle the dynamic nature of these channels effectively. This adaptability aspect could be a valuable direction for future research and development in this field. We have added this paper in the revised version of the manuscript (i.e., [30]).

5)       The explanation of Figure 5 is not rigorous enough. Mean squared error (MSE) does not decrease with the increase of Epoch all the time. The author should explain the three sudden increases in MSE while the Epoch is less than 20.

Thank you for your insightful comment. In the realm of deep learning, and specifically in the case of Generative Adversarial Networks (GANs) like ours, the convergence of the algorithm is influenced by a multitude of factors. These include the nature of the optimization problem, key parameters such as the choice of optimizer and learning rate, the quality and characteristics of the dataset, the architecture of the model, and the application of regularization techniques. GANs, in general, are known for their unstable convergence process, often encountering issues such as mode collapse and the vanishing or exploding gradient problem, as highlighted in our paper. To mitigate these challenges, we have employed the Wasserstein objective function in our approach, as detailed in [27],[34]. This function has been shown to significantly reduce the impact of these common issues, enhancing the stability and effectiveness of the GAN training process.

6)       What did the numbers next to the UAV nodes in Figure 6 mean?

Thank you for your comment. The numbers indicate the height of the UAV. For clarity, we modified the caption of Fig. 6 in the revised version of the manuscript.

7)       In the third paragraph on page 3, ‘More specifically, it addresses the …… by utilizing deep-generative models’, the meaning here is basically the same as the previous sentence. The author should check the full manuscript to avoid repeated expressions.

Thank you for your comment. The manuscript has been refined as per advised.

Comments on the Quality of English Language

Minor editing of English language required.

Thank you for your comment. The English of the paper has been revised as advised.

Round 2

Reviewer 3 Report

Comments and Suggestions for Authors

We have no further comments.

Comments on the Quality of English Language

We have no further comments.

Author Response

Comments and Suggestions for Authors

We have no further comments.

Comments on the Quality of English Language

We have no further comments.

We would like to thank the reviewer for his/her suggestions to improve the manuscript.